# Entropy-Enthalpy Compensations Fold Proteins in Precise Ways

**DOI:** 10.3390/ijms22179653

**Published:** 2021-09-06

**Authors:** Jiacheng Li, Chengyu Hou, Xiaoliang Ma, Shuai Guo, Hongchi Zhang, Liping Shi, Chenchen Liao, Bing Zheng, Lin Ye, Lin Yang, Xiaodong He

**Affiliations:** 1National Key Laboratory of Science and Technology on Advanced Composites in Special Environments, Center for Composite Materials and Structures, Harbin Institute of Technology, Harbin 150080, China; 19B918053@stu.hit.edu.cn (J.L.); maxiaoliang@hit.edu.cn (X.M.); 20B918032@stu.hit.edu.cn (S.G.); zhanghongchi@hit.edu.cn (H.Z.); shiliping@hit.edu.cn (L.S.); 2School of Electronics and Information Engineering, Harbin Institute of Technology, Harbin 150080, China; houcy@hit.edu.cn (C.H.); 1170500818@stu.hit.edu.cn (C.L.); 3Key Laboratory of Functional Inorganic Material Chemistry (Ministry of Education), School of Chemistry and Materials Science, Heilongjiang University, Harbin 150001, China; zhengbing@hlju.edu.cn; 4School of Aerospace, Mechanical and Mechatronic Engineering, The University of Sydney, Sydney, NSW 2006, Australia; lin.ye@sydney.edu.au; 5Shenzhen STRONG Advanced Materials Research Institute Co., Ltd., Shenzhen 518035, China

**Keywords:** protein-folding, entropy, enthalpy, H-bonds, thermodynamic

## Abstract

Exploring the protein-folding problem has been a longstanding challenge in molecular biology and biophysics. Intramolecular hydrogen (H)-bonds play an extremely important role in stabilizing protein structures. To form these intramolecular H-bonds, nascent unfolded polypeptide chains need to escape from hydrogen bonding with surrounding polar water molecules under the solution conditions that require entropy-enthalpy compensations, according to the Gibbs free energy equation and the change in enthalpy. Here, by analyzing the spatial layout of the side-chains of amino acid residues in experimentally determined protein structures, we reveal a protein-folding mechanism based on the entropy-enthalpy compensations that initially driven by laterally hydrophobic collapse among the side-chains of adjacent residues in the sequences of unfolded protein chains. This hydrophobic collapse promotes the formation of the H-bonds within the polypeptide backbone structures through the entropy-enthalpy compensation mechanism, enabling secondary structures and tertiary structures to fold reproducibly following explicit physical folding codes and forces. The temperature dependence of protein folding is thus attributed to the environment dependence of the conformational Gibbs free energy equation. The folding codes and forces in the amino acid sequence that dictate the formation of β-strands and α-helices can be deciphered with great accuracy through evaluation of the hydrophobic interactions among neighboring side-chains of an unfolded polypeptide from a β-strand-like thermodynamic metastable state. The folding of protein quaternary structures is found to be guided by the entropy-enthalpy compensations in between the docking sites of protein subunits according to the Gibbs free energy equation that is verified by bioinformatics analyses of a dozen structures of dimers. Protein folding is therefore guided by multistage entropy-enthalpy compensations of the system of polypeptide chains and water molecules under the solution conditions.

## 1. Introduction

Proteins are the building blocks of life on Earth and they perform a vast array of functions within organisms. Each nascent protein exists as an unfolded polypeptide when translated from a sequence of mRNA to a polypeptide chain in a ribosome. The intrinsic biological functions of a protein are determined by its native three-dimensional (3D) structure that derives from the physical process of protein folding [1], by which means a polypeptide folds into its native characteristic and functional 3D structure in an spontaneous manner. Protein folding can thus be considered the most important mechanism, principle, and motivation for biological existence, functionalization, diversity, and evolution [2,3,4].

The protein-folding problem was brought to light over 60 years ago. Given the complexity of protein folding, the protein-folding problem has been summarized in three unanswered questions [1]: (i) What is the physical folding code in the amino acid sequence that dictates the particular native 3D structure? (ii) What is the folding mechanism that enables proteins to fold so quickly? (iii) Is it possible to devise a computer algorithm to effectively predict a protein’s native structure from its amino acid sequence? A fourth essential question is: Why does protein folding highly depend on the solvent (water) [5] and the temperature [5]? Since Christian Anfinsen shared a 1972 Nobel Prize in Chemistry for his work revealing the connection between the amino acid sequence and the protein native conformation [6], understanding protein sequence-structure relationships has become the most fundamental task in molecular biology, structural biology, biophysics and biochemistry [7]. Several experimental methods are currently used to determine the structure of a protein. Around 180 million amino-acid sequences are known to science; only some 170,000 of them have had their structures experimentally determined and stored in the protein data bank (PDB) archives.

Protein folding is one of the miracles of nature that human technology finds quite difficult to follow, due to the very large number of degrees of rotational freedom in an unfolded polypeptide chain. In the 1960s, Cyrus Levinthal pointed out that the apparent contradiction between the astronomical number of possible conformations for a protein chain and the fact that proteins can fold quickly into their native structures should be regarded as a paradox, known as Levinthal’s paradox [8]. Levinthal also pointed out there should be pathways for protein folding [9]. Despite a lot of progress being made in the prediction of protein native structures through the use of artificial intelligence [10], understanding the physical folding mechanisms and laws still remains the most fundamental task in molecular biology and biophysics. As stated in Anfinsen’s Dogma, the “thermodynamic hypothesis” means that the three-dimensional structure of a native protein in its normal physiological milieu (solvent, pH, ionic strength, presence of other components such as metal ions or prosthetic groups, temperature, and other) is the one in which the Gibbs free energy of the whole system is lowest; that is, that the native conformation is determined by the totality of interatomic interactions and hence by the amino acid sequence, in a given environment [6]. The well-defined native 3D structures of small globular proteins are uniquely encoded in their primary structures (i.e., the amino acid sequences), and are kinetically reproducible and stable under a range of physiological conditions. There must be physical mechanisms that allow polypeptide chains to find the native states encoded in their sequence [1]. Protein folding can therefore be considered as an organized reaction.

In trying to solve the protein-folding problem, the concept of conformational Gibbs free energy function was introduced and enabled the modification of Anfinsen’s thermodynamic hypothesis to single molecule thermodynamic hypothesis accordingly [11,12]. The conformational Gibbs free energy function can be denoted as GX;U,E for a globular protein U in a environment E [11,12], any conformation X=x1,⋯,xn∈ℝ3n is a variables of it. Let a1,⋯, an be all atoms, then x1,⋯,xn is just their nuclear positions. Environment constitute important parameters, for example, a conformation X1 in environment E1 is a stable conformation of U and means it is a local minimizer of X;U,E1, hence ∇GX1;U,E1=0. However, in another environment E2, it is quite possible that ∇GX1;U,E2≠0, **X**_1_ cannot be a stable conformation in environment E2. Simply increasing temperature will cause U’s physiological environment EU to change to another environment E, thus the native structure XU of U will not be a stable conformation in environment E, i.e., ∇GXU;U,E≠0. This is an explanation of denaturation in different temperature, the stable conformations in environment E will be different from XU. In general, this is summarized as the single molecule thermodynamic hypothesis [13]:

In the case of a protein U in an environment E, all stable conformations XE are local or global minimizers of GX;U,E. This means that, if XU∈ℝ3n is the set of all conformations of U, there are neighbourhoods UxE∈XU of XE, such that
(1)GXE;U,E=minX∈UxEGX;U,E

Since $G\left(X;U,E\right)$ is differentiable,
(2)$$\nabla G\left(X_{E};U,E\right)=0$$

The conformation **X** is not only a single point in ℝ3n, it also represents a 3-dimensional entity occupying a region by the molecule at conformation **X**. This entity is usually modeled by the CPK model (a bunch of balls) PX=∪i=1nBxi,ri, where Bxi,ri is a round ball in ℝ3 centred at xi with the van der Waals radius ri. Since the van der Waals radius is the same for all atoms of the same element, PX is uniquely determined by **X**, and vice versa.

Moieties of a protein molecule U are classified into *l* > 1 hydrophobicity classes, such that water molecules with nearby class moiety have surface chemical potential *ω_i_*, *i* = 1, ..., *l*. Accordingly, all atoms in U can be classified into hydrophobicity classes, 1≤i≤l, such that
$$\left\{a_{1},\;a_{2},\cdots ,a_{n}\right\}=\cup_{i=1}^{k}H_{i},\;H_{i}\cap H_{j}=\emptyset ,\;i\neq j$$

In ℝ3, there are interfaces between the entity PX and the aqueous solvent, each of them is completely determined by PX. Let ΣX be one of them, for example, the largest connected branch of ∂PX, the boundary of PX. Then, ΣX bounds a bounded domain ΩX with a finite volume V (ΩX) > 0. We denote A(S) the area of a surface S.

Divide into hydrophobicity surfaces $\Sigma_{X,i}$, *i* = *i* = 1, ..., *l* as follows: Let $P_{X,i}=\cup_{a_{j}\in H_{i}}B\left(x_{i},r_{i}\right)$, then $P_{X}=\cup_{i=1}^{l}P_{X,i}$. Define
(3)$$\Sigma_{X,i}=\left\{y\in \Sigma_{X}:dist\;\left(y,P_{X,\;i}\right)\leq dist\;\left(y,P_{X,\;i}\;\backslash \cup_{j\neq \;i}P_{X,\;j}\right)\right\}$$
where $dist\;\left(y,P_{X,\;i}\right)$ is the distance from y to $P_{X,\;i}$ defined by
$$dist\;\left(y,P_{X,\;i}\;\right)=\underset{x\in P_{X,\;i}}{\min}\left|y-x\right|$$

When $E_{U}$ is the aqueous solvent, the conformational Gibbs free energy function $G\left(X;U,E_{U}\right)$ has an analytic formula derived via quantum statistics [11,12]:(4)$$G\left(X;U,E_{N}\right)=\omega_{e}V\left(\Omega_{X}\right)+\omega_{e}d_{\omega }A\left(\Sigma_{X}\right)+\sum_{i=1}^{l}\omega iA\left(\Sigma_{X,\;i}\right)+\sum_{1\leq i\leq j}^{n}\frac{Z_{i}Z_{j}e^{2}}{4\pi \epsilon \left|X_{i}-X_{j}\right|}$$
where $\omega_{e}$ > 0 is the per volume chemical potential of an electron, $Z_{i}$ is the number of protons in the atom $a_{i}$, $d_{\omega }$ is the diameter of a water molecule, and ϵ is the dielectric constant but here may be a space function.

By the hydrophobicity classification, when a water molecule closes to $\Sigma_{X,\;i}$, its per unit chemical potential is $\omega_{i}$. When $H_{i}$ is hydrophobic, $\omega_{i}$> 0, the surface repulses water; when $H_{i}$ is hydrophilic, $\omega_{i}$< 0, the surface attracts water, for example, by forming intermolecular hydrogen bonds. We can rearrange $\omega_{i}$ by decreasing order such that
(5)$$\omega_{1}>\cdots \omega_{k}>0>\cdots \omega_{l}$$

Accordingly, the interface can be decomposed into hydrophobic and hydrophilic surfaces:$$\Sigma_{X}=\left(\cup_{i=1}^{k}\Sigma_{X,\;i}\right)\cup \left(\cup_{i=k+1}^{k}\Sigma_{X,\;i}\right)=\Sigma_{X,h}\cup \Sigma_{X,p}$$
where $\Sigma_{X,h}$ is hydrophobic surface because on it $\omega_{i}$ > 0 thus it repulses water molecules and $\Sigma_{X,p}$ is hydrophilic surface, because on it $\omega_{i}$< 0 such that it attracts water molecules. Thus,
$$\sum_{i=1}^{k}\omega_{i}A\left(\Sigma_{X,\;i}\right)\;\mathrm{and}\;\sum_{i=1}^{l}\omega_{i}A\left(\Sigma_{X,\;i}\right)$$
represents the positive contribution of hydrophobic surface $\Sigma_{X,h}$ and the negative hydrophilic contribution of the hydrophilic surface $\Sigma_{X,p}$ to the Gibbs free energy $G\left(X;U,E_{U}\right)$ [11,12]. Therefore, shrinking the areas of the hydrophobic surface $\Sigma_{X,h}$ will decrease the conformational Gibbs free energy $G\left(X;U,E_{U}\right)$. Similarly, enlarging the area of hydrophilic surface $\Sigma_{X,p}$ will also decrease $G\left(X;U,E_{U}\right)$.

Anfinsen pointed out that reducing Gibbs free energy is the folding force: “This (folding) process is driven entirely by the free energy of conformation that is gained in going to the stable, native structure” [6]. Therefore, intrinsic folding force (distinguishing from friction with water and random heat vibrating) on an atom a_i_ can be expressed as:(6)$$F_{i}^{I}=-\nabla_{x_{i}}G\left(X;U,E_{U}\right),\;i=1,\cdots ,n.$$

This can be decomposed into packing force [11,12];
(7)$$F_{i}^{P}=-\nabla_{x_{i}}\left[\omega_{e}V\left(\Omega_{X}\right)+\omega_{e}d_{\omega }A\left(\Sigma_{X}\right)\right]$$
aqueous force;
(8)$$F_{i}^{A}=-\nabla_{x_{i}}\left[\sum_{i=1}^{l}\omega_{i}A\left(\Sigma_{X,i}\right)\right]$$
and expansion force.
(9)$$F_{i}^{E}=-\nabla_{x_{i}}\left[\sum_{1\leq j\neq i\;}^{n\;\;}\frac{Z_{i}Z_{j}e^{2}}{\;4\pi \epsilon \left|x_{i}-x_{j}\right|}\right]$$

In particular, the aqueous force will automatically start to shrink the hydrophobic surface $\Sigma_{X,h}$ and enlarge the hydrophilic surface $\Sigma_{X,p}$, resulting the collapse of hydrophobic moieties into a hydrophobic core that will induce the entropy-enthalpy compensation. Although the Formula 3 does not include hydrogen bonds, the mechanism of the formation of intramolecular hydrogen bonds within polypeptide chains can be revealed by the analysis of the entropy-enthalpy compensation.

There are many unresolved questions regarding the role of water in protein folding [14,15,16]. The interaction of protein surface with the surrounding water is often referred to as the protein hydration layer (also sometimes called the hydration shell) and is fundamental to the structural stability of protein, because non-aqueous solvents in general denature proteins [17]. The hydration layer around a protein has been found to have dynamics distinct from the bulk water to a distance of 1 nm, and water molecules slow down greatly when they encounter a protein [18]. For many proteins or protein domains, relatively rapid and efficient refolding can be observed in vitro. Thus, proteins should be regarded as “folding themselves” following explicit folding pathways [1]. Protein folding has been considered a free energy minimization process that is guided mainly by the following physical forces: (i) formation of intramolecular hydrogen (H)-bonds, (ii) van der Waals interactions, (iii) electrostatic interactions, (iv) hydrophobic interactions, (v) chain entropy of protein, and (vi) thermal motions [19,20]. Currently, the generally accepted hypothesis (i.e., the folding funnel hypothesis) in the field is to conceive of protein folding in a funnel-shaped energy landscape, where every possible conformation is represented by a free energy value. The rapid folding of proteins has been attributed to random thermal motions that cause conformational changes leading energetically downhill toward the native structure that corresponds to its free energy minimum under the solution conditions [1,19]. However, there are both enthalpic and entropic contributions to the free energy of proteins according to the Gibbs free energy equation that change with temperature and so give rise to heat denaturation, and in some cases, cold denaturation [21]. So far, the hypotheses have been unable to decipher the folding code; therefore, it is not generally possible to read a sequence and predict the shape it will adopt. The relationship between the folding funnel hypothesis and the Gibbs free energy equation has not been revealed in detail. The hydrophobic effect has been considered a major driving force in protein folding. The correlation between the hydrophobic effect and protein-folding mechanisms have not been clearly revealed [16,22,23].

Protein folding is highly dependent on the folding of typical secondary structures as the means to hierarchically pave a native folding pathway. Several hypotheses have been proposed to explain the folding mechanism of β-sheets. The hydrophobic zipper hypothesis proposes that a hairpin, first formed before hydrophobic contacts, acts as a constraint that brings other contacts into spatial proximity [24]. This leads to further constraints and causes the rest of the contacts to zip together. Munoz et al. proposed that the folding of a β-hairpin initiates at the turn and propagates toward the tail [25]. In particular, they found that stabilization through hydrophobic contacts between residues and hydrogen bonding interactions are important for the formation of the β-hairpin. Petrovich et al. [26] studied a 37-residue triple-stranded β-sheet protein via MD simulations. Their results indicated that a β-hairpin first appears before the third strand joins in to complete the β-sheet at the end of the folding process. They ascribed the folding mechanism of the β-sheet to the combination of an initial hydrophobic collapse and a zipper mechanism, which serve to nucleate the hairpin formation. Notably, the three cited mechanisms suggest that the folding of a β-sheet is necessarily preceded by the occurrence of a β-turn. However, we still lack a “folding mechanism” for β-sheets. By mechanism, we mean a narrative that explains how the time evolution of a β-sheet folding development derives from its amino acid sequence and the solution conditions.

## 2. Results

Protein folding is accompanied by the progressive formation of intramolecular H-bonds within polypeptide chains, so the mechanism of the formation of these intramolecular H-bonds can be considered the folding mechanism [27,28]. The covalent nature of H-bonds has been highlighted in many studies [29,30]. Therefore, protein folding can also be considered as a chemical reaction [31]. To form these intramolecular H-bonds, polypeptide chains need to escape from hydrogen bonding with the surrounding strong polar water molecules. Binding energy (hydrogen bond energy) of the H-bonds between the N-H groups and C=O groups of the main chain in secondary structures is about −3.47 kcal/mol [32,33], whereas binding energy of the H-bonds between the N-H groups and water molecules is about −7.65 kcal/mol and binding energy between the C=O groups and water molecules is about −4.7 kcal/mol [32,34,35,36,37]. Thus, the calculated ΔH for the formation of a hydrogen bond between the N-H groups and C=O groups of the main chain in water is about 2.7 kcal/mol. At constant temperature and pressure, the change in Gibbs free energy is defined as ΔG = ΔH − TΔS, where ΔH represents the change in enthalpy, and ΔS represents the change in entropy. Thus, the N-H groups and the C=O groups of the main chain cannot spontaneously escape from hydrogen bonding with the surrounding strong polar water molecules and then hydrogen bond with each other to initiate the folding without any entropy-enthalpy compensations, according to the Gibbs free energy equation. Moreover, experiments have shown that secondary structures of protein (such as -helices and -sheets) are stabilized by H-bonds between the N-H groups and C=O groups of the main chain [38,39]. The nature of polypeptide chain folding is determined by intramolecular H-bond formation between the acceptor CO and the donor NH groups [40]. This finding also indicates that the shielding effect of surrounding water molecules prevents hydrophilic side-chains from interfering with the formation of secondary structures during protein folding. Thus, water molecules should be able to saturate the H-bond formations of hydrophilic side-chains before the protein folding [18,41,42], since water molecules have a shielding effect [43,44].

As for the entropy production, water molecules tend to segregate around the “hydrophobic” side-chains of a nascent protein in an aqueous environment, creating hydration shells of ordered water molecules [45]. Ordering of water molecules around a hydrophobic region increases order in a system and thereby contributes a negative change in entropy (less entropy in the system) [46]. The water molecules are fixed in these water cages that drive the hydrophobic collapse or the aggregation of the hydrophobic groups. Hydrophobic domains of an unfolded protein constrain the possible configurations of surrounding water, and so their collapse upon folding increases the water’s entropy. By aggregating the hydrophobic regions, the solvent can reduce the surface area exposed to non-polar side-chains, thus reducing localized areas of decreased entropy [47]. Experimental results show that water molecules slow down greatly when they encounter hydrophobic areas of a protein, and the speed is reduced about by 99% [18]. Thus, the standard molar entropy of water within the ordered cages around the nonpolar surface (i.e., hydration shell) is approximately equal to the standard molar entropy of solid water, and that is about 45 J mol^−1^K^−1^ [48], whereas the standard molar entropy of liquid water is about 70 J mol^−1^K^−1^ [48]. Considering that the spatial layouts of the side-chains on the typical secondary structures of α-helices and β-sheets always have the hydrophobic side-chains laterally clustered together on their surfaces, the folding increases the entropy via the laterally hydrophobic collapse of these hydrophobic side-chains (see Figure 1) [49]. For example, when two hydrophobic side-chains of leucines residues are laterally collapsed together, they can approximately expel about 12 ordered water molecules from the hydration shell of the side-chains into liquid water solvent [18]. Thereby, at room temperature (298 K), the calculated change in Gibbs free energy is about −23 kcal/mol due to the leucine–leucine hydrophobic interaction, which absolute value is obviously bigger than the calculated ΔH requirement for formation of an intramolecular H-bond in the main-chain. Thus, laterally hydrophobic collapse of two hydrophobic side-chains is capable of providing the entropy-enthalpy compensation for the formation of an intramolecular H-bond within the main-chain of the polypeptide, as long as the hydrophobic collapse and the formation of an intramolecular H-bond are coordinated in the molecular structure. Thereby, the correlation between the folding funnel hypothesis and the Gibbs free energy equation should be the entropy-enthalpy compensation at local structures of polypeptide chains in the water environment.

The entropy-enthalpy compensation should be indispensable for the spontaneous folding of a protein [50]. This explains why intrinsically disordered proteins (IDPs) and regions (IDRs) exist and make up a significant part of the proteome: the entropy-enthalpy compensation is absent in these IDPs and IDRs [51]. The constant competition among surrounding water molecules, neighboring C=O groups and N-H groups of adjacent peptide planes of the main-chain in the formation of H-bonds most likely keeps the single bonds of the backbones of IDPs and IDRs rotating randomly due to lack of hydrophobic residues [51]. The early steps of protein folding should be not directly dominated by the formation of intramolecular H-bonds, due to the shielding effect of surrounding water molecules (i.e., the ΔH calculation) and the Gibbs free energy equation. Thus, this problem may lie in our lack of understanding of how hydrophilic groups of polypeptides can escape from hydrogen bonding with the surrounding strong polar water molecules at early steps of the folding, given the lack of awareness of the importance of the Gibbs free energy equation in governing the formation of the H-bonds. Moreover, hydrophilic side-chains of proteins are also normally hydrogen-bonded with surrounding water molecules in aqueous environments, thereby preventing the surface hydrophilic side-chains of proteins from randomly hydrogen bonding together [18,41,42]. This is the reason why proteins usually do not aggregate or crystallize in unsaturated aqueous solutions [52], even though the solvent-facing surface of the proteins is usually composed of predominantly hydrophilic regions.

Experimental methods such as laser temperature jumping technology and single molecule experimental techniques have revealed that protein folding first leads to the formation of secondary structures (α-helices and β-strands); the tertiary structure is formed by folding of the secondary structures [31]. Therefore, protein folding is highly dependent on the folding of secondary structures as the means to hierarchically pave a native folding pathway that leads to the formation of correct tertiary structures. Thus, deciphering the physical folding codes in the amino acid sequences that dictate the formation of typical secondary structures should be regarded as a key to cracking the protein-folding problem. It is most likely that the nascent polypeptide forms an initial local secondary structure through minimization of the hydrophobic portions of neighbored side-chains in sequence exposed to water due to the hydrophobic effect [53], and thereby they create localized regions of predominantly clustered hydrophobic side-chains. Then, the secondary structure interacts with water, thereby inducing thermodynamic pressure on those regions, which then aggregate or “collapse” into a tertiary conformation with a hydrophobic core [53]. Among secondary structural types in proteins, the β-sheet and the α-helix are the most prevalent. The folding codes that dictate the formation of β-strands and α-helices should be encoded in the amino acid sequence of these segments of polypeptide chains. A β-sheet consists of β-strands connected laterally by backbone H-bonds, forming a generally pleated sheet. A β-strand is a stretch of polypeptide chain with a backbone in an extended conformation. The folding of unfolded polypeptide segments into the β-strands is most likely earlier than the folding of β-sheets, it is very difficult to explain how the lateral hydrogen bonding process of segments of unfolded polypeptide (i.e., the folding process of a β-sheet) is simultaneously accompanied by the stretching process of polypeptide chain segments into these β-strands. Therefore, there should be entropy-enthalpy compensations that allow polypeptide chain segments to find the states of β-strands encoded in their sequence. In typical β-strands, each carbonyl oxygen atom (C=O) in a peptide plane is hydrogen bonding with an amide hydrogen atom (N-H) in an adjacent peptide plane due to the electrostatic attractions between them, causing a tendency for the C=O group and N-H group to be parallel to each other, namely, the parallel distribution of adjacent peptide planes (see Figure 1). The feature of parallel distribution of adjacent peptide planes and thereby parallel distribution between each side-chain and every other side-chain is prevails in almost all experimentally determined β-strands (see Figure 1). There must be entropy-enthalpy compensations promote these hydrogen bonding in-between the C=O groups and N-H groups of the backbones of β-strands due to the Gibbs free energy equation [43,44]. Note that the structure of β-strands not only cause the parallel distribution of adjacent peptide planes but also make adjacent side-chains to distribute on opposite sides of the main chain and each side-chain is parallel to every other side-chain, enabling hydrophobic proportions of the neighbored side-chains laterally hydrophobic collapse together (see Figure 1). The hydrophobic interaction among the neighboring side-chains of β-strands can reintroduce entropy to the system via the breaking of their water cages which frees the ordered water molecules [54]. The hydrophobic collapse between each side-chain and every other side-chain in β-strands initiates the enthalpy-entropy compensation, enabling the adjacent peptide planes to bond with each other through hydrogen-bonding between the neighbored C=O groups and N-H groups as shown in β-strands (see Figure 1). Thus, all single bonds in the backbone of the unfolded polypeptides should be rotationally hindered in the β-strand-like thermodynamically metastable state. The entropy-enthalpy compensation should be responsible for the formation of β-strands due to the Gibbs free energy equation and the molecular configuration. Experimental evidence for the folding of unfolded proteins provides corroboration for a hypothesis that sites of folding initiation arise from hydrophobic interactions [20,22,53].

## 3. Discussion

It has previously been noted that many amino acid side-chains contain considerable nonpolar sections, even if they also contain polar or charged groups [22,55]. That is, hydrophilic side-chains are not entirely hydrophilic. The hydrophilicity of hydrophilic side-chains is normally expressed by CO or NH groups at their ends, whereas the other portions of hydrophilic side-chains are hydrophobic, because the molecular structures of these portions are basically alkyl and benzene ring structures, as shown in Figure 2. Therefore, the folding initiation sites of secondary structures might contain not only accepted “hydrophobic” amino acids, but also long hydrophilic side-chains [22]. The hydrophobic portions of the hydrophilic side-chains are most likely involved in the laterally hydrophobic interaction among neighbored side-chains for secondary structures formation. Cysteine-C, Isoleucine-I, Leucine-L, Methionine-M, Tryptophan-W, Phenylalanine-F, Tyrosine-Y, and Valine-V can be fully involved in hydrophobic interaction with adjacent hydrophobic side-chains due to their high hydrophobicity (see Figure 2a). Arginine-R, Histidine-H, Lysine-K, Glutamate-E and Glutamine-Q also can actively become involved in hydrophobic interaction with adjacent hydrophobic side-chains in sequence, due to their long hydrophilic side-chains contain long nonpolar alkyl structures, (see Figure 2b). Aspartate-D and Asparagine-N would permit very limited participation in hydrophobic interaction with neighboring side-chains in sequence because their exposed hydrophobic proportions are relatively small (see Figure 2c). Alanine-A most likely can laterally hydrophobic attract with long hydrophilic side-chains, due to its hydrophobic side-chain is short enough to hydrophobic attract with hydrophobic proportions of these hydrophilic side-chains and without repelling with the hydrophilic tops of these side-chains (see Figure 2d). Glycine-G cannot effectively participate in lateral hydrophobic interaction with other neighbored side-chains in folding of a β-strand, because the hydrophobic proportion of its side-chain is negligible (see Figure 2e). Note that Proline-P normally cannot directly contribute to the formation of β-strands through the entropy-enthalpy compensation, because Proline-P does not contain the N-H group in the main-chain (see Figure 2f) that causes no H-bond formation between adjacent peptide planes at the residue of the backbone (see Figure 1). Thus, Proline-P normally terminate β-strands formation. When a hydrophobic side-chain can avoid latterly approaching to the hydrophilic proportion of a hydrophilic side-chain, we can conceive that the hydrophobic side-chain can laterally hydrophobic attract the hydrophilic side-chain, as a method for predicting whether a hydrophilic side-chain can laterally hydrophobic attract another hydrophobic or hydrophilic side-chain.

Since the formation of β-strands is driven by hydrophobic interactions among neighboring side-chains of unfolded polypeptide in sequence and guided by the enthalpy-entropy compensation according to the Gibbs free energy equation [54], we should be able to find experimental evidence of the hydrophobic interaction in the PDB archives. We use 1000 experimentally determined small protein structures to demonstrate and verify the hydrophobic-effect-based folding mechanism in β-sheets (see Appendix A). All the 1000 small proteins were randomly selected from the PDB. Among them, α-type proteins accounted for 27.3%, β-type proteins accounted for 14.3%, α/β-type proteins accounted for 2.9%, and α+β-type proteins accounted for 55.5%. There are 45 similar sequences in the 1000 samples. With use of the PDB archive and the STRIDE software [56], 3427 typical β-strands (four or more amino acids long) can be identified in the 1000 protein structures. From analysis of all the 3427 β-strands of the 1000 proteins in the PDB, we find that the phenomenon of hydrophobic side-chains or hydrophobic portions of the hydrophilic side-chains latterly clustering together (due to the hydrophobic effect) on one side or the other of β-strands is prevalent in all experimentally determined β-sheets. This finding confirmed that the hydrophobic interactions among neighboring side-chains and the entropy-enthalpy compensations are responsible for the formation of β-strands. Hydrophobic effects can contribute to the formation of β-sheets through multistage aggregations of neighboring hydrophobic groups of unfolded polypeptides and the entropy-enthalpy compensations, leading to the formation of β-strands that subsequently fold into β-sheets (see Figure 3).

A de novo designed protein (PBDID: 5TPJ) is a good example to illustrate the phenomenon of hydrophobic attraction (due to the hydrophobic effect) among adjacent side-chains on each β-strand of a protein (see Figure 4) [57]. To illustrate the hydrophobic attraction, we highlight the hydrophobic surface areas of adjacent side-chains on each β-strand of the protein, based on the experimentally determined protein structure, as shown in Figure 4c,d. Note that every β-strand is characterized by a large hydrophobic surface fully covering one side of the β-brand (the inner side), and causing each side-chain to be parallel to every other side-chain of each strand, due to the hydrophobic interaction. Parallel distribution of neighboring “hydrophobic” side-chains in a β-strand can effectively reintroduce entropy to the system via the merging of the water cages of the side-chains, which frees the ordered water molecules (see Figure 4d). Thus, the β-strand should be considered an initial metastable state for many unfolded polypeptide segments corresponding to its free energy minimum under the solution conditions, creating localized regions of predominantly hydrophobic proportions of side-chains [41]. Lateral hydrogen bonding process of segments of β-strands during the folding process of a β-sheet should be also driven by hydrophobic interactions among the side-chains and entropy-enthalpy compensations, as shown in Figure 3. β-sheets folding highly depends on the temperature [5], where β-sheets can form in as little as one microsecond after a temperature jump [58,59,60]. The temperature dependence of folding of β-sheets is thus attributed to the temperature dependence of the Gibbs free energy equation.

The β-turn is the third most important secondary structure after helices and β-strands. Aspartate-D, Asparagine-N, Serine-S, and Glycine-G cannot effectively hydrophobic attract with neighboring side-chains in sequence because the hydrophobic proportions of their side-chains are very small (see Figure 2). Proline-P normally cannot directly contribute to the formation of β-strands through the entropy-enthalpy compensation, since Proline-P does not contain the N-H group in the main-chain. Thus, Aspartate-D, Asparagine-N, Serine-S, Proline-P, and Glycine-G most likely lead to the formation of β-turns in protein folding, due to the tendency of the other neighboring hydrophobic side-chains in the amino acid sequence to hydrophobically collapse together by bypassing these residues. β-turns have been classified in accordance with the values of the dihedral angles φ and ψ of the central residue. β-turns can easily be identified between β-strands or α-helices of protein structures using the PDB archive and the STRIDE software [56]. We identified 5776 β-turns in the 1000 protein structures, including about 1780 β-hairpin turns. We found that about 97.4% of the β-turns contained at least one Aspartate-D, Asparagine-N, Serine-S, Proline-P or Glycine-G residue [61], as illustrated in Appendix A. Moreover, about 99.3% of β-hairpin turns contain at least one residue of Aspartate-D, Asparagine-N, Serine-S, Proline-P or Glycine-G (see Appendix A).

We use another small-molecule protein (PBDID:1OUR) as an example, to demonstrate the role played by hydrophobic interactions among neighboring side-chains in the formation of β-strands, β-turns, and β-sheets (see Figure 5). The protein is mainly comprised of β-strands and 10 β-turns. Every β-strand of the protein is also characterized by a large hydrophobic surface fully covering one side of the β-strand (see Figure 5a). Aspartate-D, Asparagine-N, Serine-S, Proline-P, Glycine-G contribute to the formation of β-turns in protein folding, because the other neighboring side-chains in the β-strands tend to hydrophobically attract to each other through bypassing these residues (see Figure 2). Thus, Aspartate-D, Asparagine-N, Serine-S, Proline-P, and Glycine-G can be classified as a hydrophobic blocking (RB) group. It is worth noting that almost all the 10 β-turns of the protein are composed with two or more residues of Aspartate-D, Asparagine-N, Serine-S, Proline-P, Glycine-G (see Figure 5a,b). This indicates that two or more adjacent RB residues can effectively block hydrophobic attraction among neighboring side-chains in sequence on both sides of a strand. We plot the protein structure in three parts in accordance with three segments of the amino acid sequence to illustrate the hydrophobic collapse among neighboring β-strands in sequence (see Figure 5b,c). Hydrophobic interactions among these β-strands cause them to collapse together through bending the unfolded polypeptide at the location of these RB residues. This observation also indicates that the entropy-enthalpy compensations drive hydrophobic attraction and hydrogen bonding among the β-strands to fold into the β-sheets. The formation of β-sheets also causes the β-strands to aggregate or “collapse” into a tertiary conformation with a hydrophobic core. Thereby, the folding of β-sheets is triggered by multistage hydrophobic interactions and entropy-enthalpy compensations among neighboring residues of unfolded polypeptides, enabling β-sheets to fold following explicit physical folding codes (see Figure 3, Figure 4 and Figure 5).

There should be entropy-enthalpy compensations that allow polypeptide chain segments to find the states of α-helices encoded in their sequence. An α-helix structure usually has a large number of hydrophobic side-chains agglomerated on its surface (see Figure 6). The folding of the α-helix structure may be also driven by the hydrophobic collapse of adjacent side-chains in the sequence through the entropy-enthalpy compensations. The typical state of a β-strand is that each residue side-chain can directly hydrophobic interact with the two adjacent residue side-chains at 1 interval in the sequence, as shown in Figure 1 and Figure 3. The side-chain of each residue in the α-helix structure can have a hydrophobic interaction with the surrounding four residue side-chains at two or three intervals in the sequence (see Figure 6a), which means that the entropy value of some polypeptide segments in forming the α-helices can be higher than that in forming the β-sheets. Therefore, the formation of the α-helix can be regarded as a further entropy-enthalpy compensation of the polypeptide segment from the β-strand-like thermodynamic metastable structure. The formation of α-helices enable laterally hydrophobic collapse among these side-chains of residues at two and three intervals in the amino acid sequence (see Figure 6). Therefore, when the amino acid sequence of a polypeptide fragment not only meets the structural requirements for β-strand, but also can have strong lateral hydrophobic interaction among the residues at three or three intervals in the sequence, it cause the polypeptide segment to fold into an α-helix instead of a β-strand. If a post-translational modification changes the critical lateral hydrophobic interactions among the residues at two or three intervals in the sequence, the polypeptide segment will most likely not fold into the α-helix due to the absence of the critical hydrophobic forces.

The tertiary structure of an arabidopsis protein (PDBID: 1Q4R) is a composed of typical secondary structures and is suitable as a simple example to illustrate how the entropy-enthalpy compensation mechanism can be used to predict the secondary and tertiary structures. We summarized the basic laws of laterally hydrophobic attraction and hydrophobic repulsion between side-chains of different residues. The rules of hydrophobic interaction among the side-chains of adjacent residues in the polypeptide chain sequence that causes the folding of α-helix and β-sheet are initially explored. When a fragment of a polypeptide chain in the β-strand-like thermodynamically metastable state shows sufficient hydrophobic attraction between the side-chains of adjacent residues on one side, it can be predicted that the fragment will fold into a β-strand or an α-helix. When the fragment also satisfies that a strong hydrophobic attraction can occur among the residues at two and three intervals in the sequence, it can be predicted that the polypeptide fragment will fold into an α-helix instead of a β-strand. The entropy-enthalpy compensation analysis of the amino acid sequence fragment of the protein 1Q4R is illustrated in Figure 7.

The results show that the folding codes in the amino acid sequence that dictate the formation of β-strands, α-helices and turns can be deciphered through the evaluation of the hydrophobic interactions among neighbored side-chains of an unfolded polypeptide from a β-strand-like thermodynamic metastable state with great accuracy of prediction. The folding process of a tertiary structure from secondary structures is also involved in the entropy-enthalpy compensation mechanism, since a β-sheet structure can be regarded as a partial tertiary structure. Six other examples are illustrated in Appendix A. The folding of secondary structures make hydrophobic side-chains cluster together, thereby inducing thermodynamic pressure on neighbored secondary structures in sequences, which then aggregate or “collapse” into one or more global conformations with one or more hydrophobic cores. This explains why multi-domain proteins sometimes have multiple hydrophobic cores. Enthalpy-entropy compensation may allow some secondary structures folding on the ribosome as this allows certain order of folding of local hydrophobic cores of different domains.

In order to prove that the entropy-enthalpy compensation mechanism is the protein-folding mechanism and can be used to predict the secondary structure of proteins, we preliminarily program a simple software (See Appendix A) for predicting the typical secondary structures of α-helices and β-sheets based on the entropy-enthalpy compensation analysis of the amino acid sequences (https://www.researchgate.net/publication/353445795_software, accessed on 30 July 2021) similar to that shown in Figure 7 and Appendix A. Using this software, we successfully identified 5837 of the samples are basically β-strands and α-helices, covering about 96 percent of all those β-strands and 92 percent α-helices in the 1000 proteins (see Appendix A). Only 0.5% samples are neither β-strands not α-helices. Hydrophobic effects can most likely contribute to the formation of α-helices through implementing the hydrophobic interaction among neighbored side-chains two or three residues intervals. We used this to identify α-helices from these samples. Then, we identified 2308 samples of β-strands of three or more amino acids long, making the successful rate of the prediction about 81%. We also identified 2416 samples of α-helices, making the successful rate of the prediction about 87% (see Appendix A). Moreover, physical folding codes for β-strand and α-helices can be quickly deciphered by using the software, making the overall time for prediction for the 1000 proteins less than 30 s by using only one CPU. We used another 1000 experimentally determined small protein structures to test the software. There were 188 similar sequences in the 1000 samples. All the 1000 small proteins were also randomly selected from the PDB. By using the software, we identified 5915 of the samples are basically β-strands and α-helices, covering about 93 percent of all those β-strands and α-helices in the 1000 proteins. Another 327 samples (about 0.5%) are false predictions. The successful rate of the prediction for β-strand is about 80% and the successful rate of the prediction for α-helix is about 86% (see Appendix A). Lateral hydrogen bonding process of segments of β-strands during the folding process of a β-sheet is driven by hydrophobic interactions among β-strands and therefore the entropy-enthalpy compensations (see Figure 3 and Figure 4). Thus, a large β-sheet structure can be regarded as a partial tertiary structure. Our model directly predicted the secondary structures in full-length, that is, different from the assembly pathway captured by the molecular dynamics trajectories (see Appendix A) [62]. By analyzing these 2000 proteins, we found that hydrophobic amino acids account for about 55% of the amino acids in the β-strands, and hydrophobic amino acids account for about 47% of the amino acids in the α-helices. About 95% hydrophobic side-chains in the β-strands are involved in hydrophobic interaction with other hydrophobic side-chains in the secondary structures. About 96% hydrophobic side-chains in the α-helices are involved in hydrophobic interaction with other hydrophobic side-chains in the secondary structures.

The assembling process of tertiary structures into a quaternary structure is likely to be essentially the same as that of protein docking. A recent theoretical study found that the binding affinity between the cellular receptor human angiotensin converting enzyme 2 (ACE2) and receptor-binding domain (RBD) in spike (S) protein of novel severe acute respiratory syndrome coronavirus 2(SARS-CoV-2) is determined by the hydrophobic interaction between them [55]. The hydrophobic interaction and enthalpy-entropy compensation in the binding region between the S protein and ACE2 protein enable the hydrophilic residues in this region to discard the hydrogen-bonded water molecules, and to promote intermolecular hydrogen bonding and electrostatic attraction among these hydrophilic side-chains at the binding site [55]. Therefore, the folding of protein quaternary structures should be guided by the entropy-enthalpy compensations in between the docking sites according to the Gibbs free energy equation. Namely, entropy increments caused by hydrophobic surface areas collapse in-between protein subunits compensate the increment of enthalpy caused by H-bonds formation between protein subunits. The distribution of hydrophobic and hydrophilic surface areas at smooth docking sites can be easily analyzed from their projective images (see Figure 8). Through analyzing the hydrophobic attraction relationships among proteins of hundreds of dimeric proteins, we find out that the docking position of a dimer is always characterized by two rules of the distribution of hydrophobic and hydrophilic surface areas in their projective images of the overlapping map. First, the docking position maximizes the overlapping of hydrophobic surface areas of the two projective images of the protein subunits. Secondly, subunit–subunit docking sites must allow several hydrogen bond donors and acceptors close to each other in the overlapping position of the two projective images, enabling the formation of several H-bonds between them. Obviously, these two rules conform to the theory that the entropy-enthalpy compensation dominates subunit–subunit docking of dimers into quaternary structures. We had programmed a simple software (https://www.researchgate.net/publication/352552505_software, accessed on 30 July 2021) by using the two rules for predicting the docking position between two projective images of a protein–protein complex. To prove that the folding process from subunit structures into quaternary structures is guided by the entropy-enthalpy compensations, we try to predict the overlapping position of the docking sites of 12 dimers in two dimensions of the projective images (see Figure 8 and Appendix A) by using this software and the two rules of the entropy-enthalpy compensation at the interfaces. By using the software, we find out that the docking position between two projective images of a dimer can be accurate predicted through rotation and translation of the two projective images following the two rules. All the overlapping positions of the docking sites of 12 dimers in two dimensions were successfully predicted by the using the software, which provides potent proof for the entropy-enthalpy compensation theory. All the 12 dimers have relatively smooth binding sites and were randomly selected from the PDB. The docking position between subunit structures indeed maximize the hydrophobic collapse of hydrophobic surface areas of the binding sites in-between the protein subunits.

## 4. Materials and Methods

### 4.1. Protein Structures

In this study, many experimentally determined native structures of proteins were used to study the folding mechanisms of β-sheets. All the three-dimensional (3D) structural data of protein molecules were resourced from the PDB database. The IDs of these proteins according to the PDB database are marked in the figures. To illustrate the distribution of hydrophobic areas on the surface of β-strands and β-sheets in these figures, we used the structural biology visualization software PyMOL to display the hydrophobic surface areas of these secondary structures.

### 4.2. Identification of Secondary Structures of Proteins

The secondary structures of β-strands, β-turns, β-sheets, and α-helices were identified in the 2000 proteins using the STRIDE software [56]. We also used the molecular 3D structure display software PyMOL to confirm the identification of the secondary structures of proteins.

## 5. Conclusions

The core of the protein-folding problem is to crack the folding mechanism. By mechanism, we mean a narrative that explains how the time evolution of secondary and tertiary structures’ folding development derives from its amino acid sequence and solution conditions. The folding mechanism must be able to explain the differences and similarities of different protein-folding pathways. The folding mechanism must also be able to explain why the secondary structure of the protein is formed first, and the more global tertiary structure is formed after the formation of the secondary structure. The folding mechanism also be able to explain why protein folding highly depends on the water solvent and the temperature. The hydrophobic interaction among the neighboring side-chains of the polypeptide chain and the thereby multistage entropy-enthalpy compensation mechanism drive the entire folding process throughout the developments of secondary, multi-domain, tertiary, and quaternary structures that can clearly explain the accuracy of protein folding in time sequence. H-bonds play an extremely important role in stabilizing protein structures. To form these H-bonds, polypeptide chains need to escape from hydrogen bonding with the surrounding strong polar water molecules in aqueous environments that require entropy-enthalpy compensation according to the Gibbs free energy equation and the calculated change in enthalpy. The folding codes and forces in the amino acid sequence that dictate the formation of secondary structures can be deciphered through evaluation of the hydrophobic interactions among neighbored side-chains of an unfolded polypeptide from a β-strand-like thermodynamic metastable state with great accuracy of the prediction. The multistage entropy-enthalpy compensations of polypeptide chains and surrounding water molecules are the folding mechanisms, enabling proteins to fold reproducibly and quickly, following explicit physical folding codes in aqueous environments. Our preliminary results show that computer algorithms based on logic judgement of entropy-enthalpy compensation relationships among neighbored residues of an unfolded polypeptide from the β-strand-like thermodynamic metastable state can be devised to effectively and accurately predict a protein’s native secondary structures from its amino acid sequence. This is little different from the methods of energetically searching for the global minimum in a given protein’s energy landscape that employed by these successful artificial intelligence algorithms [10,63]. The temperature dependence and water solvent dependence of protein folding are thus attributed to the requirements of the entropy-enthalpy compensations according to the Gibbs free energy equation. The β-strand-like thermodynamically metastable states of unfolded proteins/polypeptides most likely can be experimentally observed by using solution nuclear magnetic resonance (NMR) method at low temperature. The folding of protein quaternary structure is guided by entropy-enthalpy compensations at the docking sites in between protein subunits according to the Gibbs free energy equation that is verified by bioinformatics analyses of a dozen structures of dimers.

The folding funnel hypothesis and the proposed folding theory of entropy-enthalpy compensations mutually confirm each other to a certain extent. The folding funnel hypothesis essentially says “the Gibbs free energy formula can describe protein folding”. The funnel-shaped energy landscape theory also emphasizes that the entropy change and enthalpy change in the process of protein folding are the key factors that drive protein folding. We demonstrate that the detail steps in the protein folding processes of secondary, tertiary and quaternary structures strictly conform to the Gibbs free energy equation in the local space of the solution conditions, that is, the formation of H-bonds in the protein-folding processes satisfies the entropy-enthalpy compensation requirements for the spontaneous reaction in the local space of the solution conditions. This guarantees that the protein-folding problem can be solved by using the second law of thermodynamics, even without using any artificial intelligence algorithm. The hydrophobic environment of a chaperon may contribute to initiate some critical entropy-enthalpy compensations encoded in the amino acid sequence, enable proteins folding into different stabilized conformations in different environments.

## Figures and Tables

**Figure 1 ijms-22-09653-f001:**
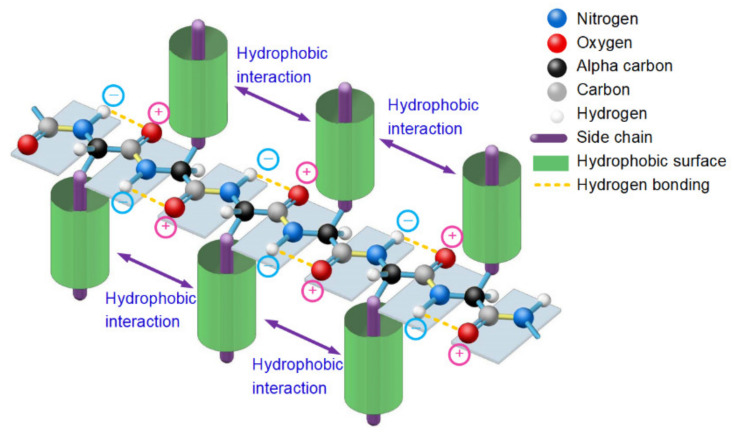
A thermodynamically metastable state of unfolded proteins is the parallel distributed state of adjacent peptide planes due to hydrophobic interactions among neighbored side-chains and the hydrogen bonding between each carbonyl oxygen atom and adjacent amide hydrogen atom in peptide plane and the entropy-enthalpy compensation, as with a typical β-strand.

**Figure 2 ijms-22-09653-f002:**
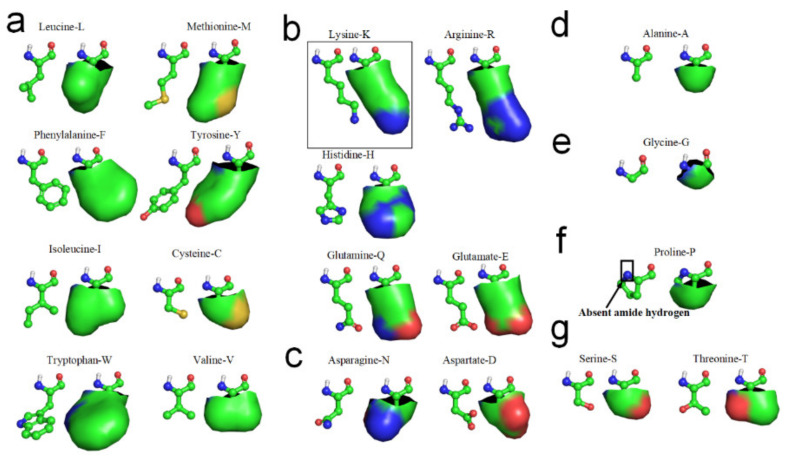
Hydrophobic portions of amino acid side-chains (hydrophobic portions are highlighted green). (**a**) Leucine, Methionine, Phenylalanine, Tyrosine, Isoleucine, Cysteine, Tryptophan, Valine. (**b**) Lysine, Arginine, Histidne, Glutamine, Glutamate. (**c**) Aspartate, Asparagine. (**d**) Alanine. (**e**) Glycine. (**f**) Proline. (**g**) Serine, Threonine.

**Figure 3 ijms-22-09653-f003:**
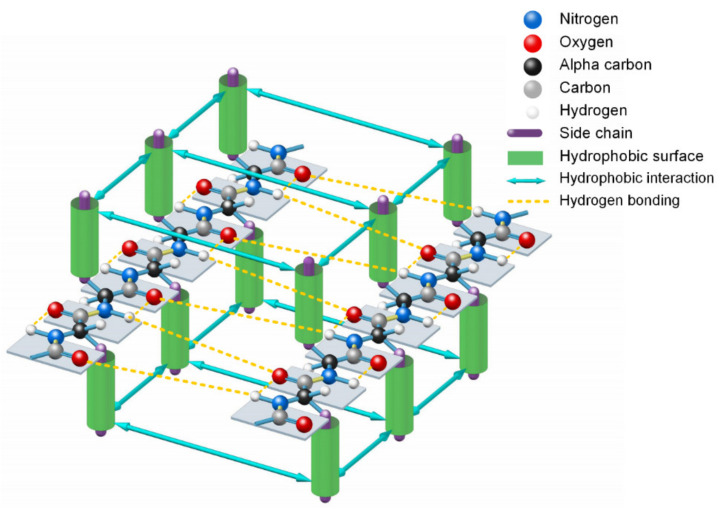
Lateral hydrogen bonding process of segments of two β-strands in folding a β-sheet driven by hydrophobic interactions among side-chains and entropy-enthalpy compensations.

**Figure 4 ijms-22-09653-f004:**
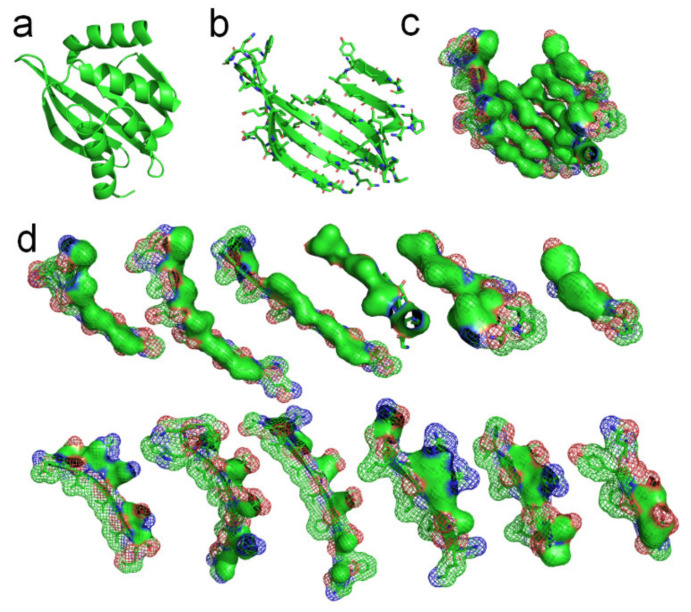
Hydrophobic attraction among neighboring side-chains of β-strands. (**a**) A de novo designed protein (PBDID: 5TPJ). (**b**) The curved β-sheet of 5TPJ. (**c**) Hydrophobic attraction among adjacent β-strands via the hydrophobic surfaces of side-chains of the β-sheet (hydrophobic surfaces are highlighted green). (**d**) Hydrophobic surface areas on the 6 β-strands of the sheet (green areas).

**Figure 5 ijms-22-09653-f005:**
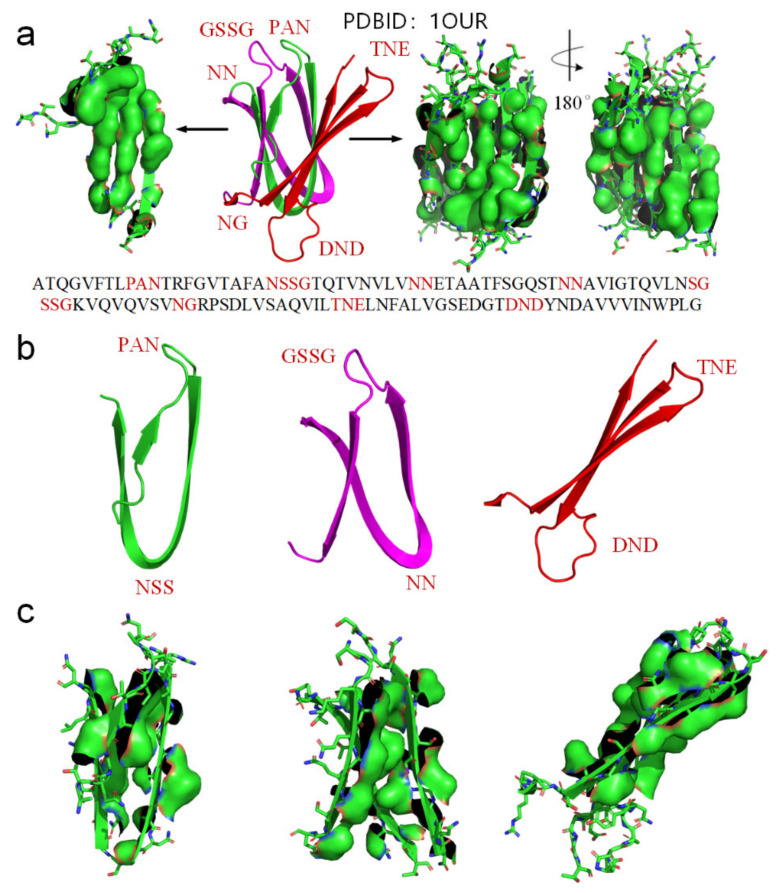
(**a**) Hydrophobic surface areas on the β-strands of the protein (PDBID: 1OUR), hydrophobic surface of side-chains is highlighted by green surface areas, residues located at turns are highlighted red in the protein sequence. (**b**) The parts of the protein (residues 1–33 highlighted green, residues 34–71 highlighted magenta, residues 72–114 highlighted red). (**c**) Hydrophobic surface areas on the β-strands of the sheet (green surface areas).

**Figure 6 ijms-22-09653-f006:**
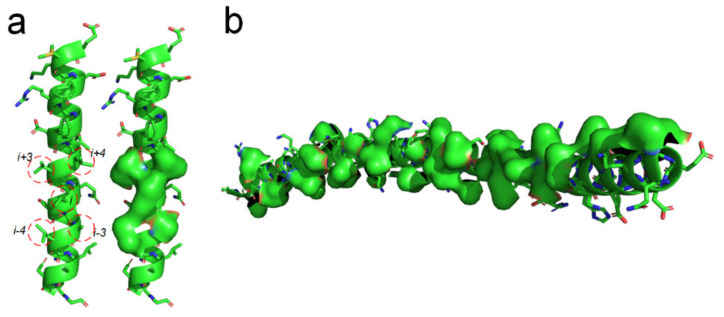
Lateral hydrophobic attraction among neighbored side-chains on α-helices. (**a**) Strong hydrophobic interaction among side-chains of the residues at 2 and 3 intervals in the amino acid sequence of a α-helix (PBDID: 5YM7); (**b**) A long α-helix with a long hydrophobic surface area on it caused by the hydrophobic side-chain distribution (PBDID: 2BEZ).

**Figure 7 ijms-22-09653-f007:**
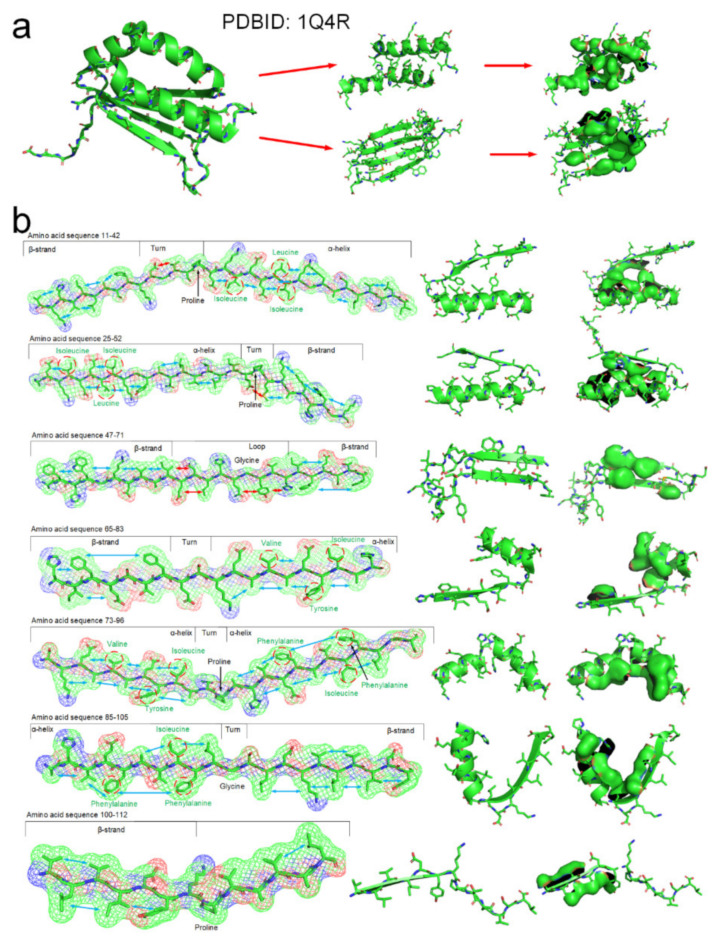
The folding mechanism of a protein structure (PBDID: 1Q4R) based on entropy-enthalpy compensation. (**a**) Hydrophobic interaction among side-chains of secondary structures. (**b**) The polypeptide chain fragment and the corresponding secondary structure in a thermodynamically metastable state are drawn in 7 segments (the hydrophobic attraction between the side-chains of adjacent residues is marked with a blue arrow, and the hydrophilic-hydrophobic repulsion is marked with a red arrow). The proline and glycine that led to the formation of the corner structure are marked. The hydrophobic amino acids in the sequence that cause the metastable collapse to form an α-helix structure are annotated by red circles.

**Figure 8 ijms-22-09653-f008:**
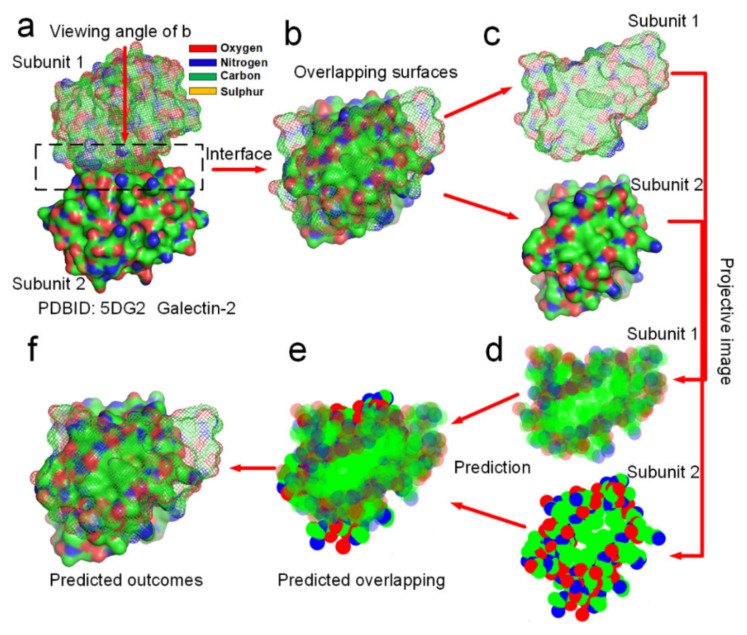
Prediction of the docking position between two protein subunits of the galectin-2 dimer in two dimensions by using entropy-enthalpy compensation mechanism. (**a**) The galectin-2 dimer. (**b**) Distribution of hydrophobic (green areas) and hydrophilic (red and blue areas) surface areas on the two protein subunits at the docking site. (**c**,**d**) Projective images of distribution of hydrophobic and hydrophilic surface areas at the binding site. (**e**) The predicted maximized the overlapping of hydrophobic surface areas of the two projective images of the two protein subunits. (**f**) The prediction of the docking position between the two protein subunits in two dimensions, almost same as (**b**).

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
