# Peer review of "Entropy-Enthalpy Compensations Fold Proteins in Precise Ways"

_ijms, 2021, doi:10.3390/ijms22179653_

Round 1

Reviewer 1 Report

Li et al present a manuscript detailing the importance of entropy-enthalpy compensation in the folding landscape of proteins. The manuscript is clear and concise but has a few issues.

  • Statistical analysis of the data is missing. Authors must provide statistical analysis of their analysis of the PDB subset. E.g. how many similar sequences are in the data set? what are the probability of b-strands for different amino acid pairs? What are the probabilities of hydrophilic vs hydrophobic amino acid pairs appearing in strands? Etc.
  • Why did authors exclude helices from their analysis? If the entropy-enthalpy compensation truly is the key for folding, as the authors claim, then helices should also show the same tendencies; no?
  • Since the PDB structures are not of dynamics folding processes, the authors should test their model by using the available simulation data from DE Shaw which actually shows the folding dynamics of several proteins from different classes.
  • In light of the successes of both the AlphaFold2 and RoseTTaFold it would be prudent to have a discussion of how/to what extent the findings of the manuscript are reflected in those algorithms (which place emphasis on sequence and amino acid interactions).
  • The authors need to discuss multi-domain. In particular, the multi-domain proteins that fold on the ribosome and whose active conformations must fold in step-wise fashion.
  • Authors must also discuss the roles of post-translational modifications and chaperons. These are important factor for folding in vivo. Folding in the hydrophobic environment of a chaperon does not carry the same energetic penalty as folding in aqueous environment. Structure determination studies introduce unnatural, and sometimes improbable, chemical environments to obtain crystals or stabilized conformations for NMR. I understand that a simplistic representation must be used to establish a baseline model, nevertheless critical factors must be mentioned – potentially discussed as future prospects.
  • Missing Figure 12.

Minor issues:

  • English grammar needs a bit of polish.
  • L113-125: subscript formatting and missing items.
  • Did authors use stride_ss in PyMOL? Otherwise they need to, to keep consistent with use of STRIDE.
  • Software associated with the manuscript should be uploaded as Supplement.
  • Please do not use RAR archives. ZIP or TAR is supported on all software platforms.

Reviewer 2 Report

The discussion of protein structure generation based on the H-bonds system analysed using entalpy-entropy relation.

The paper is correct.

However I do not see any progress in protein structure prediction and analysis.

The about 40 lines long Conclusion does not give any novelty in problem treating.

Not all available models for protein folding problem are mentioned in the paper. It is impossible to discuss all models applied, however few available based on hydrophobicity could be mentioned.

The number of examples very limited.

The proteins taken as examples of the length < 130 aa with high content of secondary structure.

Please use the protein 2L42 as the example.

Round 2

Reviewer 1 Report

- The authors have performed rudimentary statistical analysis of the secondary structure distribution. However, they still do not show the statistics for hydrophobic vs hydrophilic residues in secondary structure elements, or distribution of amino acid pairs that arise when e.g. a two-strand sheet is formed. E.g. for a b-strand that is 15 amino acids long, how many residues are hydrophobic? They have the data in their 2000 samples, it just needs to be analyzed.

- Authors misunderstood my point with DE Shaw trajectories. Their model should not predict how dynamic assembly occurs but authors should compare how their predictions match with the assembly pathway captured by the molecular dynamics trajectories available from DE Shaw. E.g. do their predictions align with the formation of secondary structure at certain segments? Are the predicted secondary structure elements the same as those captured by DE Shaw simulations? Etc.

- Multi-domain proteins: The authors have not discussed if their model is usable for multi-domain proteins. E.g. some protein must fold on the ribosome as this allows certain order of folding for the domains. Re-folding such proteins in vitro, without the ribosome, does not give an active protein. The question now is, can the model of enthalpy-entropy compensation predict or describe these phenomena?

- Post-translational modifications are crucial entities for a large subset of proteins. Authors did not discuss how their model may be affected by these changes to the amino acids.

Round 3

Reviewer 1 Report

Authors have responded to the critiques points.